# Numerical Investigation of the Influence of Aerodynamic Loads on the Resonant Frequency of a Compressor Blade Made of EI-961 Alloy

**DOI:** 10.3390/ma15238391

**Published:** 2022-11-25

**Authors:** Arkadiusz Bednarz, Krzysztof Puchała, Michał Sałaciński, Volodymyr Hutsaylyuk

**Affiliations:** 1Faculty of Mechanical Engineering and Aeronautics, Rzeszow University of Technology, Ave. Powstancow Warszawy 12, 35-959 Rzeszow, Poland; 2Faculty of Mechanical Engineering, Military University of Technology, 00-908 Warsaw, Poland; 3Air Force Institute of Technology, Airworthiness Division, 01-494 Warsaw, Poland

**Keywords:** resonance, jet engine, strength of material, modal, FSI, loads

## Abstract

The aim of this work was to numerically determine the influence of aerodynamic loads on the value of the resonant frequency of the compressor blade. The object of the research was the 1st stage compressor blade of the PZL-10W engine. As part of the research, analytical calculations of the resonance frequency were performed and compared with the literature ones (first, second, and third forms of forced vibrations). In the next step of the investigation, a computational model of the compressor stage (fluid domain and rotors) was built and FSI analysis was performed. This analysis was based on CFD modeling of the state of aerodynamic loads on the blade surfaces, and then these values were imported as external loads for the structural analysis, which was the basis for the modal analysis, in which the resonant frequency of the first three vibration modes was determined. As part of the analyzes, both the influence of aerodynamic loads and the rotational speed of the compressor rotor were verified. Thus, it was possible to evaluate the influence of both the rotational speed (and the arising centrifugal force) and the influence of the emerging aerodynamic load. The results obtained will allow for the assessment of the impact of the aforementioned operating conditions of the aircraft engine on the resonance frequency, which in turn may translate into the durability of critical components of the aircraft engine.

## 1. Introduction

Due to the fact that the blades of the compressor of aircraft engines are critical components of a turbine engine [1,2], they are subject to rigorous testing and many regulations [3]. This is due to the working conditions of the blade, that is, the relatively high rotational speed together with the influence of aerodynamic loads [4]. Additionally, in the event of an imbalance, vibrations can occur of both the individual elements and the entire rotor [5,6]. There are many potential causes of vibration occurrence [7], from unbalance to uneven combustion. The need to protect the rotors against operation under resonance conditions is crucial for the proper operation of the turbine engine.

Running compressors can pick up hard objects from the ground/airfield and not related to the apron. Sucking them into the engine can cause damage to the leading edges [8]. In the case of the operation of a turbine engine with damaged blades, in the area of resonant frequencies, the damage itself may cause the blade to break quickly and tear it off. Such an event [9,10] was the cause of many aircraft failures.

One of the ways to counteract the operation of the blades close to resonance conditions [11] was to properly test them and determine their resonance frequencies. In addition, the appropriate modeling of the blade allows you to increase the stiffness and minimize the chance of resonance or increase the frequency at which it occurs. There was a lot of research on the resonance frequencies of blades. These were mainly experimental studies [12,13,14] in which the blade was permanently mounted on the head of the device inducing the resonance state. The tests concerned both ordinary rotor blades [13,14] and those with special protective coatings [12]. Unfortunately, such studies do not provide information on how operating conditions will affect the resonance frequency. Due to the development of technology and the possibility of numerical calculations, it is possible to estimate the resonance frequency of the blade with increasing rotational speed. There were also works [15,16] that deal with the flow of aerodynamic loads in the operation of the compressor stage. Unfortunately, there was still a lack of knowledge on the influence of aerodynamic loads on stiffness, which directly affects the frequency of resonant vibrations.

Fluid-structure interaction (FSI) is an analysis in which the behavior of a fluid/air in a known flow channel is first modeled [17,18,19], and then the obtained values of the pressure distributions (later called aerodynamic loads [17,18]) are imported [19] as a structural load acting on the subject object. This was the structure of the analysis that will be presented in this article.

To prepare a flow analysis, it is necessary to define the operating conditions of a given flow machine [20], as well as to prepare the geometry of the flow channel, hereinafter referred to as the fluid domain. Moreover, it is important to properly select the boundary conditions and the turbulence model. A similar approach should be used when the strength analysis is performed [21], which will include loads from the aforementioned flow analysis. The research will result in an evaluation of the influence of aerodynamic loads on the resonance frequencies of the turbine engine compressor blades.

The aim of this study is to determine how aerodynamic loads and rotor speed affect the resonant frequencies of the axial compressor of the selected turbine engine. It was necessary to take into account the rotational speed in the analyzes because it generates the volumetric loads (centrifugal force), which account for the greater part of the loads applied to the blade. In addition, due to the nature of the operation of the turbine engine and the rotor of the axial compressor, the rotary motion creates a suction force and adds kinetic energy to the flowing medium, therefore, when taking into account the aerodynamic loads (i.e., air pressure), the rotational speed itself should also be taken into account. The results of the conducted research will be the value of the resonance frequency of the first three modes of natural vibrations, the evaluation of the convergence of the results of numerical analyzes with the experiment, as well as the assessment of the possibility of using analytical calculations in estimating the resonance frequency of elements with complex geometry. The development and research work on engines are crucial matter from cost and ecological point of view. Various materials and solutions are tested. Different materials behaves differently under the same conditions so the exact state of loads and their influence of construction made of selected material seems to be interesting. The scope of the research carried out includes analytical calculations, the construction of a blade geometric model, numerical flow analysis, numerical strength analysis, and modal analysis of the compressor blade.

## 2. Resonance Frequency Calculation

### 2.1. Object of the Study

According to the assumptions of the work, the object of the research is the axial compressor blade of the first stage PZL-10W turbine engine. The blade is made of the EI-961 alloy [8]. The geometry of the tested blade is presented in Figure 1. The engine itself is used to drive the W3 Sokół helicopter [22].

The tested blade weighs an average of 15.6 g [23] and was subjected to surface treatment (peening), which causes residual stresses on the surface of the blade, with values locally exceeding −400 MPa (compressive stresses), significantly increasing the fatigue strength of the blade in question [24]. Similar solutions have been used globally in the structure of the rotor blades, also known as the critical elements of the engine.

### 2.2. Literature Data

The first stage blade of the PZL-10W compressor has been the subject of many experimental and numerical studies [8,23,24,25]. As part of experimental research, its fatigue life was tested depending on the damage or its absence. One of the works also focused on determining the resonance frequencies for several modes of natural vibrations and their shape/form [25]. Detailed data on the fatigue of the discussed material can be found in the literature [26,27].

The aforementioned research was carried out at Rzeszow University of Technology, in the Rotor Machine Dynamics Laboratory and are described in [25]. As part of the work, the resonance frequencies were estimated for the blade mounted horizontally in the vibration exciter using a vibrating system (fixing surface presented at Figure 1. These values are quoted in Table 1. The direction of the override is the direction of the Y axis (defined in the Figure 1).

The research carried out shows that the first mode of free vibrations, a purely bending form, occurred at a frequency of about 770 Hz [25]. The second form of vibrations was observed at a frequency of about 2445 Hz, and it was the bending-torsional form. The third form of vibrations occurred at a vibration frequency of about 3600 Hz. The third form of forced vibrations was also a bending-torsion form. It should be remembered that the cited work do not take into account rotational speed or aerodynamic loads. The quoted results are equivalent (von Mises criterion) to the numerical analysis for the case where the rotational speed 0 and the aerodynamic loads are not taken into account.

### 2.3. Analytical Calculations

To prepare the initial verification of the results of the numerical analyzes, analytical calculations were carried out to determine the first three resonance frequencies of the beam [28] with the geometry corresponding to the compressor blade. Furthermore, it will be possible to evaluate the use of this method in the preliminary estimation of resonance frequencies before performing experimental tests.

The basic formula for the resonant frequency of a beam (Equation 1) depends on the material data (Young’s modulus and material density) and geometry (cross-sectional area, moment of inertia, and beam length). An additional quantity appearing in the formula is the value of the root of the excitation equation for a given vibration mode.
(1)$$f_{res}=\frac{\alpha_{n}^{2}}{2\Pi l^{2}}\times \sqrt{\frac{EJ}{\rho A}},Hz$$
where: $\alpha_{n}$—*n*-th root of the excitation equation (where *n* represents the number of the resonance frequency);

l=53 mm—length of the beam (blade);E=210 GPa—Young’s modulus;ρ=7850 kg/m${}^{3}$—material density;A=39.0567 mm${}^{2}$—cross-sectional area (at the foot of the blade);J=22.7 mm${}^{4}$—section moment of inertia.

The values of all three resonance frequencies are summarized in Table 2. The frequency of the first form of vibrations estimated in this way was about 785 Hz, and the second over 4900 Hz. The last estimated form of vibrations III was nearly 14,000 Hz.

The estimated values will be used later in the work of comparing the results. They will also be used as an initial method to verify the accuracy of the data obtained through numerical analyzes.

## 3. Numerical Analysis

The last method used to determine resonance frequencies was numerical analysis. The numerical analyzes themselves were conducted in two ways. In the first version, the analysis will be purely structural (denoted as SP analysis).

In the first type of analysis structural and modal calculation were performed. In the second one aerodynamic loads were also taken into consideration (denoted as SAL analysis) to determine resonance frequencies (Figure 2). The general assumptions of the analysis were as follows: for the geometric model of the blade, all degrees of freedom were obtained on the side surfaces of the blade root. Then, loads from spinning (rotational speed) were applied, due to which it was possible to obtain a state of loads in the blade caused by spinning. This state was the basis for further calculations based on the modal analysis, which made it possible to determine the resonance frequencies for different rotational speeds.

Coincidentally (for each rotational speed) Fluid-Structure Interaction (FSI) analysis, i.e., the structure-flow analysis through the rotor stage of the axial compressor of the turbine engine was performed. The structure of this analysis was presented in Figure 2. The purpose of this analysis was to determine the state of the aerodynamic loads that appear on the side surfaces of the blade.

Structural analyzes were performed with the usage of Ansys ver. 2019 R2 software (Ansys Inc., Canonsburg, PA, USA). Details of the above analyses will be presented in later chapters.

### 3.1. Fluid Flow Analysis

As mentioned above, in part of the numerical tests, aerodynamic loads resulting from the influx of the working medium - air will be used. For this purpose, a numerical flow analysis was prepared based on the RANS (Reynolds-average Navier-Stokes equation) method [29], which was one of the basic solvers built into Fluent commercial software.

To perform the flow analysis mentioned above, it was necessary to prepare the geometry of the fluid domain, which corresponds to the volume in which air moves through the compressor rotor channel. The created domain is presented in Figure 3. The domain was extended to include part of the inlet and outlet channels, assuming 2.5 times the blade chord. This is to stabilize the flow streams. An analysis requires the user to assume boundary conditions and operating conditions. The analyzes were conducted with the assumption of an incompressible fluid.

The assumed boundary conditions for the analysis are as follows:At the inlet—static pressure 101.325 hPa;On the inlet—speed 98 m/s (corresponding to the flight speed of the aircraft);At the outlet—the outlet was realized with a mass flow of 201 g/s (in case of 22,490 RPM—for other analysis was calculated linearly);Domain—rotational with the assumed rotor speed (variable value in the analysis);Working medium—compressible gas—air with parameters corresponding to the reference atmosphere [30].

The geometrical model of the domain was discretized using the Ansys Mesher tool. The volume size was assumed to be 2 mm, or its compaction in the vicinity of the blade surface (with the maximum imposed size of 0.2 mm). The generation of an inflation layer consisting of five layers of volumetric elements was used on the blade surfaces (Figure 3). Periodic conditions were assumed on the side surfaces of the domain, assuming the continuity of the flow between them (which simulates the operation of the entire rotor).

3 different rotational velocities of the domain, i.e., 11,000, 22,490 and 31,486 RPM were considered in the flow analyzes. The value of 22,490 RPM corresponds to the rotational speed of the drive turbine [22], while 31,486 RPM was the maximum rotational speed of the generator turbine. The result of these analyzes is the pressure distribution on the blade surfaces, as well as the velocity and pressure distribution in the analyzed domain. For better visualization of the obtained results, distributions of total pressure (Figure 4) and velocity (Figure 5) were prepared in the plane that crosses the flow channel. It can be seen that the highest value of total pressure was observed on the inside of the blade.

The pressure values obtained were imported into the flow analysis, according to the diagram in Figure 2.

### 3.2. Structural Analysis

The last part of the numerical tests was structural static analysis, taking into account the influence of the change in rotational speed and the aerodynamic loads that appear in the state of stresses and displacements in the compressor blade (SAL analysis). In addition, the obtained stress state was the basis for further modal analyzes, in which the first three frequencies of resonant vibrations were determined for given load states.

For the purpose of the strength and modal analysis, a discrete model was prepared. It was assumed that the maximum size of the finite element used to build a discrete model cannot exceed 0.2 mm. The resulting discrete model, consisting only of 10-node tetrahedral elements (SOLID187 elements with quadratic square functions), was shown in Figure 6.

In the structural analysis, degrees of freedom were obtained by blocking displacements and rotations in all nodes on the side walls of the blade lock (Figure 6)—the method of blocking degrees of freedom and the direction of the excitation correspond to the conditions of the experiment—fixing surface and Y axis on Figure 1. The aerodynamic loads (imported from flow analysis) were then modeled and applied to the side surfaces of the blade. The analysis prepared in this way, taking into account the various rotational speeds mentioned above, made it possible to determine the state of stress depending on the set rotational speed. To better illustrate the impact of rotational speed and emerging aerodynamic loads, two Table 3 and Table 4 have been prepared. The effect of both rotational speed and aerodynamic loads is presented and discussed.

The analyzes show that with increasing rotational speed, the blade deflection (maximum displacement) and the maximum value of equivalent stress (according to Huber-Mises-Hencky commonly known as von Mises criterion) $\sigma_{eqv}$ increase. In the absence of rotational speed (the blades were standing), the incoming air itself causes the blade to deflection by nearly 0.004 mm and the appearance of stresses of approximately $\sigma_{eqv}=44$ MPa. Operation at a rotational speed of 11,000 RPM increases the stress reduced to 115 MPa, and taking into account the air inflow increases their value by 44 MPa. In the mentioned case, the deflection of the blade varies from 0.3 mm to 0.45 mm (50% difference between cases). Increasing the rotational speed to the maximum, that is, 31,486 RPM, and taking into account the influx of the working medium, causes the appearance of stresses of 984 MPa and deflection of the blade by 2.62 mm.

### 3.3. Modal Analysis

In the next step modal analyzes were carried out to determine the frequency of resonant vibrations for the first three forms of forced vibrations. Their values were summarized in Table 3 and Table 4. In the case of SAL analysis the frequency of the first mode of natural vibrations, in the absence of rotational speed, was about 789 Hz, and the influx of the medium itself does not have a significant influence on its value. The shape presented in Figure 7. The first one is purely of bending mode/form. The second form occurs with a frequency of about 2520 Hz (bending-torsional form), and the third form around 3620 Hz (bending-torsional form—Figure 7). The increase in rotation speed to 11,000 RPM caused the increase in vibration form I to a value of approximately 841 Hz. A further increase in the rotational speed, to the value of 22,490 RPM, increased the resonance frequency of the I form of vibrations to approximately 985 Hz, and for the rotational speed of 31,486 RPM, this frequency was 1137 Hz. The differences between experimental and analytical for modes II and III are as a result of simplifications of analytical model and fact that they are the bending-torsional modes.

For better illustration of the effect of rotational speed and taking into account aerodynamic loads, Table 5 was prepared showing the difference between the individual results from Table 3 and Table 4. The calculations show that with increasing rotational speed, the impact of aerodynamic loads on the frequency of resonance vibrations decreases. This observation was correct for all three vibration modes analyzed.

## 4. Results Comparison

The last step of the investigation was to compare the results of the numerical, analytical and experimental analyzes. The results were summarized in Table 6. The results of the analyzes for the case of rotational speed equal to 0 (without centrifugation), the resonance frequency of the first form of vibration, estimated by analytical methods, was 784 Hz, and it was overestimated only by 14 Hz in relation to the experimental result. Numerical tests showed a higher value—around 789 Hz. The error was approximately 2.5% of the experimental value. Other forms of vibration were characterized by a similar error.

The frequency of the II mode of vibrations estimated numerically was higher than the experimental data by approximately 3% (about 76 Hz). In the case of the next form of vibration, the frequency has a very small error (about 40 Hz for analyzes with the influx of the working medium and about 2 Hz for the analysis without aerodynamic loads). The results of analytical calculations for modes II and III gave values far from the experimental data.

## 5. Conclusions

Different materials behaves differently under the same conditions, therefore the exact state of loads and their influence of construction made of selected material is of great value. Research and development work on aerospace engines are crucial matter from cost and ecological point of view, therefore, various materials and solutions are tested. This paper presents the study of the effect of rotational of aerodynamic loads on resonant frequencies of a compressor blade made of EI-961 Alloy. This effect is impossible to determine experimentally. Additionally there are no data on this subject in the literature. Therefore, the numerical calculation were used, namely, FEM and FSI ones. In the first step analytical calculations were performed to determine three first resonant frequencies. Subsequently the adequate FSI and FEM model were build and analyses were performed. In the first version, the analysis was purely structural (denoted as SP analysis). In the first type of analysis structural and modal calculation were performed. In the second one aerodynamic loads were also taken into consideration (denoted as SAL analysis). Structural analyzes were performed with the usage of Ansys ver. 2019 R2 software. Numerical flow analysis was prepared based on the RANS (Reynolds-average Navier-Stokes equation) method, which was one of the basic solvers built into Fluent commercial software. The results of the numerical calculations were compared with those of the experimental tests and the analytical calculations. On the basis of the above, it was possible to draw the following conclusions:The first form of resonant vibrations, for the tested compressor blade, occurs at about 770 Hz.The result of analytical calculations gives a satisfactory value only for the first form of resonance vibrations.The first of resonant vibrations one is purely of bending mode/form. The second and third form are bending-torsional ones.Numerical calculations show circa 3% agreement with the experimental data.In the case of the third mode of natural vibrations, the resonance frequency changes by about 40 Hz, taking into account the aerodynamic loads.The influence of aerodynamic loads on the resonance frequencies of the compressor blade decreases with increasing of the rotational speed.Aerodynamic loads increase the deflection of the blade and increase the value of reduced stresses, which does not have a significant effect on the resonance of the blade.The change in the resonant frequency was related to the change in the geometry of the blade due to the action of aerodynamic loads (greater deflection and rotation of the blade causes a minimal change in the resonance frequency).

In general, it can be stated that aerodynamic loads do not have a large effect on the frequency of the resonant vibrations. An increase in rotational speed causes an increase in centrifugal forces, which reduces the influence of aerodynamic loads (and the resulting deflection of the blade). Nevertheless, as the development of aerospace engines is ongoing and the fact that influence of analyzed parameter is impossible to verify experimentally and there are no data concerning it, the calculations and results presented in this paper are valuable.

## Figures and Tables

**Figure 1 materials-15-08391-f001:**
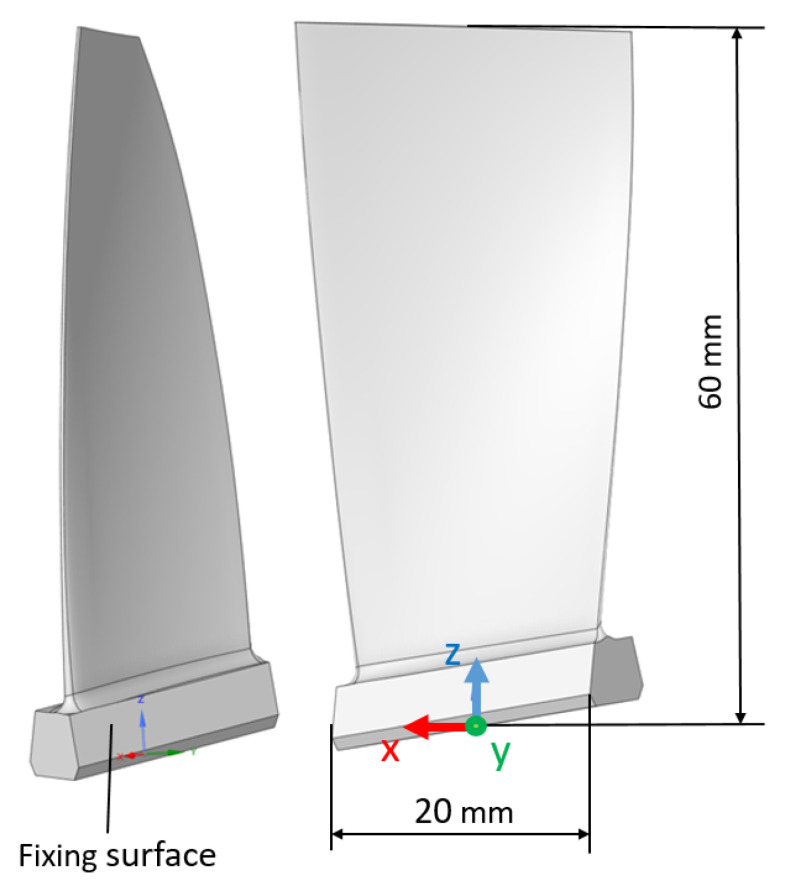
Geometrical model of the compressor blade of the first stage of the PZL-10W engine.

**Figure 2 materials-15-08391-f002:**
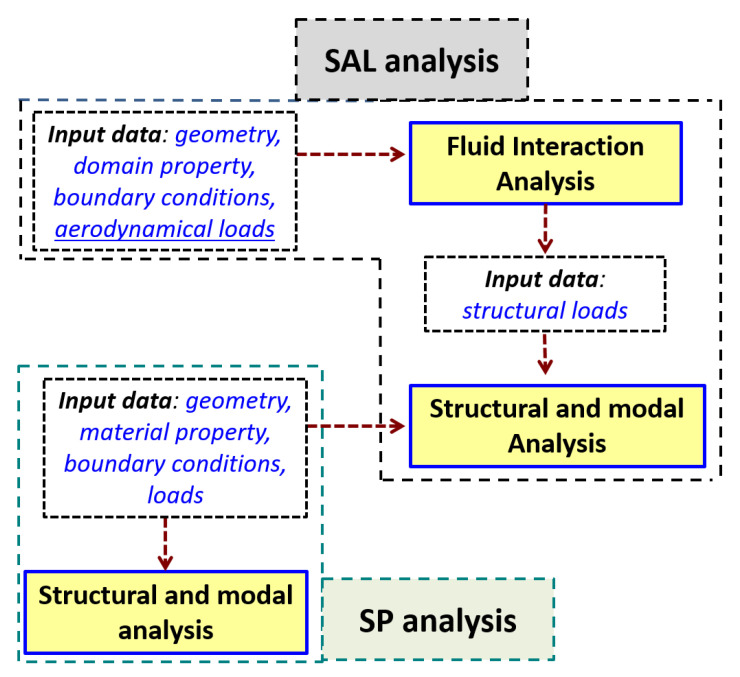
Structure of the SP (with no aerodynamic loads consideration) and SAL analysis Fluid Structure Interaction).

**Figure 3 materials-15-08391-f003:**
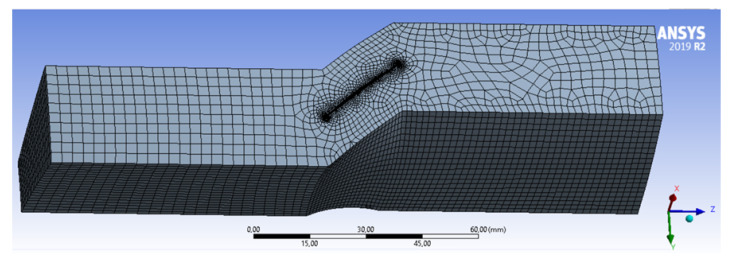
Discrete fluid domain mesh.

**Figure 4 materials-15-08391-f004:**
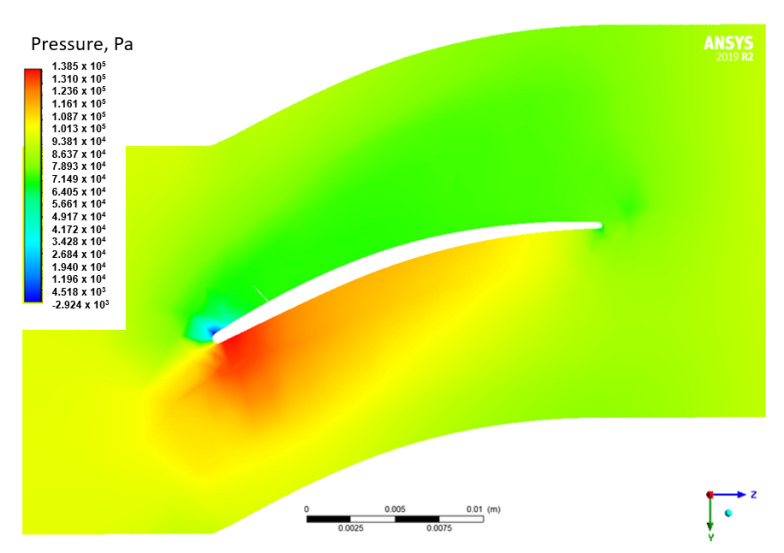
Relative static pressure distribution in the cross-section connecting the fluid domain for the rotational speed of 31,486 RPM (cross-section in 50% of the height of the blade −76 mm from the axis of rotation).

**Figure 5 materials-15-08391-f005:**
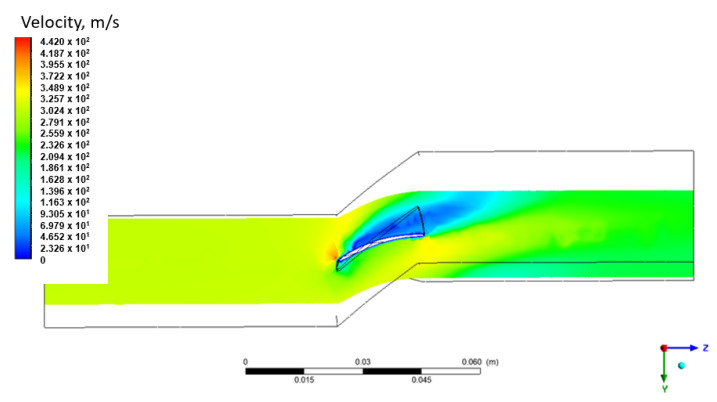
The distribution of the relative velocity (taking into account the rotational speed) in the cross section that covers the fluid domain for the rotational speed of 31,486 RPM (cross-section in 50% of the height of the blade −76 mm from the axis of rotation).

**Figure 6 materials-15-08391-f006:**
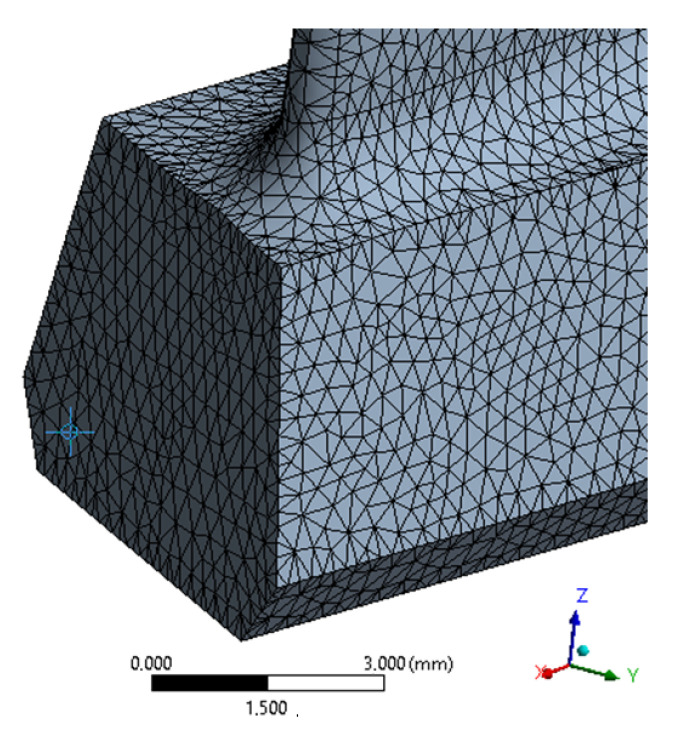
Discrete model for strength analysis.

**Figure 7 materials-15-08391-f007:**
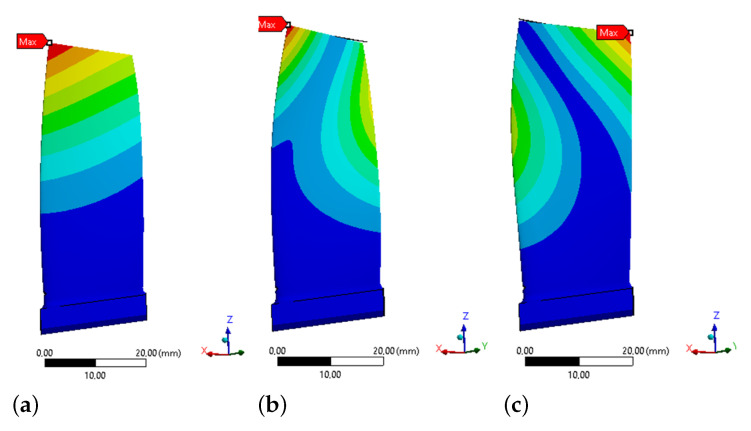
Free vibration modes of the blade: (**a**) I-st, (**b**) II-nd and (**c**) III-rd mode (SAL analysis) (results scaled to 1 mm deformation, where blue means 0 mm and red means 1 mm).

**Table 1 materials-15-08391-t001:** List of the resonant frequencies of the compressor blade of the PZL-10W engine, experimentally estimated [25].

	Resonance Frequency $f_{\mathrm{res}},\mathrm{Hz}$	
**I-st Mode**	**II-nd Mode**	**III-rd Mode**
770.31	2445.31	3600

**Table 2 materials-15-08391-t002:** Summary of the results of the calculated frequencies of resonant vibrations of the beam with the geometry convergent with the compressor blade.

	Resonance Frequency $f_{\mathrm{res}},\mathrm{Hz}$	
**I-st Mode**	**II-nd Mode**	**III-rd Mode**
784.85	4918.83	13,756.74

**Table 3 materials-15-08391-t003:** Summary of the results of numerical SP analysis (with no aerodynamic loads consideration).

Rotational Velocity, RPM	Eq. Stress $\sigma_{\mathrm{eqv}}$, MPa	Max. Deformation, mm	I-st Mode, Hz	II-nd Mode, Hz	III-rd Mode, Hz
0	0	0	789.39	2518	3598.238
11,000	114.7966	0.301005	841.2594	2556.473	3678.684
22,490	479.8689	1.258249	985.3016	2665.851	3807.039
31,486	940.543	2.466169	1136.907	2784.595	3968.938

**Table 4 materials-15-08391-t004:** Summary of the results of the numerical SAL analysis (with aerodynamic loads considered).

Rotational Velocity, RPM	Eq. Stress $\sigma_{\mathrm{eqv}}$, MPa	Max. Deformation, mm	I-st Mode, Hz	II-nd Mode, Hz	III-rd Mode, Hz
0	44.18	0.00379	789.62	2521.6	3640.3
11,000	158.9653	0.452119	841.4552	2559.462	3680.356
22,490	524.0332	1.409334	985.4381	2668.793	3808.517
31,486	984.7064	2.617247	1137	2787.488	3970.206

**Table 5 materials-15-08391-t005:** Summary of differences between SP and SAL analyzes for different rotational speeds of the compressor vane.

	I-st Mode, Hz	II-nd Mode, Hz	III-rd Mode, Hz
0	−0.23	−3.6	−42.0621
11,000	−0.19583	−2.98917	−1.6714
22,490	−0.13644	−2.94283	−1.47802
31,486	−0.09243	−2.89317	−1.26754

**Table 6 materials-15-08391-t006:** List of resonant frequencies of an axial compressor blade of a turbine engine, estimated numerically and experimentally.

	I-st Mode	II-nd Mode	III-rd Mode
Experiment	770.31 Hz	2445.31 Hz	3600 Hz
SP analysis (with no aerodynamic loads consideration)	789.39 Hz	2518 Hz	3598.24 Hz
Error to exp.	−19.08 Hz	−72.69 Hz	1.76 Hz
Percentage difference to exp.	−2.48%	−2.970%	0.049%
SAL analysis (with aerodynamic loads considered)	789.62 Hz	2521.6 Hz	3640.3 Hz
Error to exp.	−19.31 Hz	−76.29 Hz	−40.3 Hz
Percentage difference to exp.	−2.51%	−3.12%	−1.12%

## Data Availability

All data generated or analysed during this study are included in this published article.

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
