# Peer review of "Numerical Investigation of the Influence of Aerodynamic Loads on the Resonant Frequency of a Compressor Blade Made of EI-961 Alloy"

_materials, 2022, doi:10.3390/ma15238391_

Round 1

Reviewer 1 Report

This paper aimed to investigate the aerodynamic loads on influence of the resonant frequency of the compressor blade, but from the full text it is more like a modal analysis rather than the resonant analysis. And this kind of modal analysis is a basic operation before the blade design. So the characteristic of this article is not obvious and the logic is not clear. It needs a major revision before the next step. The main problems are briefly described as follows.

In the abstract, the theoretical description language is too long. On the contrary, the conclusion is basically absent. I have no idea about the innovation of this paper and contribution to research content according to the abstract. It has little relationship with the theme of the journal materials.

In the introduction, it leads a misunderstanding about the resonant analysis on the random foreign object impact on the basis of the second paragraph. But it is just a modal analysis. And it lacks a literature research on resonance analysis. Also the lack of innovation is a pity of this paper.

As the main body of the section 3 and 4, the modeling process is the most basic pre-processing for the calculation of rotating turbomachinery, especially for a single channel with the commercial software. Therefore, the focus should be on the result analysis rather than single channel modeling itself. Meanwhile, the structural and modal analysis are the premise of blade design. For resonance theoretical analysis, Campbell diagram has given a comprehensive risk prediction, which is considered the factor such as rotating speed. The influence of steady aerodynamic force on modal frequency is relatively small compared with centrifugal forc. So the dynamic aerodynamic force should be focused instead.

The conclusion with original opinions is not shown in the final part. The statements in the conclusion are common sense descriptions. The authors need to draw more original conclusions.

Author Response

Dear Reviewer,

In the beginning, I would like to thank you very much for the time spent on reviewing our manuscript and providing suggestions how to improve our publication.

All formatting errors and typos have been corrected. All suggestions were helpful and valuable. The responses and comments to individual remarks will be provided below.

The following structure was adopted in the response: first, the reviewer's remark is quoted, and then the answer is provided.

Quote: „but from the full text it is more like a modal analysis rather than the resonant analysis”.

Answer: According to the authors, the results of modal analysis are identical to the analysis of resonance frequencies (in a limited frequency spectrum).

Quote: „And this kind of modal analysis is a basic operation before the blade design. So the characteristic of this article is not obvious and the logic is not clear.”

Answer: The purpose of the article is not to design the engine but to study the effect of aerodynamic loads on resonant frequencies of a compressor blade made of EI-961 Alloy. This effect is impossible to determine experimentally so the numerical calculation were used. Additionally there are no data on this subject in the literature which was stated in the article. Additionally, an extra statement was added in Introduction chapter.

Quote: „In the abstract, the theoretical description language is too long. I have no idea about the innovation of this paper and contribution to research content according to the abstract. It has little relationship with the theme of the journal “materials”.”

Answer: In the authors’ opinion the introduction gives the necessary background to the article which was also pointed out by one of the reviewer. The innovation in this article is to show the effect of rotational speed and modeling of aerodynamic loads on resonance frequencies, using the example of a specific aircraft engine and a specific material. Conclusions were rewritten.

Quote: “In the introduction, it leads a misunderstanding about the resonant analysis on the random foreign object impact on the basis of the second paragraph. But it is just a modal analysis.”

Answer: The mention of FOD is intended to make the recipient aware that a potential damage at the leading edge may contribute to a drastic reduction in the durability of the object. The FOD itself is not a research object.

Quote: “And it lacks a literature research on resonance analysis. Also the lack of innovation is a pity of this paper.”

Answer: What reviewer means by resonant analysis? Reviewer stated that introduction is too long already.

Quote: „As the main body of the section 3 and 4, the modeling process is the most basic pre-processing for the calculation of rotating turbomachinery, especially for a single channel with the commercial software. Therefore, the focus should be on the result analysis rather than single channel modeling itself.”

Answer: In authors opinion the section 3 is necessary. The information presented  there are crucial if someone would like to check results. In this section also results are presented and commented.  In section 4 there is only results analysis and their comparison.

Quote: “For resonance theoretical analysis, Campbell diagram has given a comprehensive risk prediction, which is considered the factor such as rotating speed.”

Answer: The authors are aware of the use of the Campbell diagram at the engine design stage. Thanks to the conducted research, it was possible to clearly define whether taking into account only the rotational speed (without aerodynamic loads) is sufficient and what is the impact of the aerodynamic loads themselves.

Quote: “The influence of steady aerodynamic force on modal frequency is relatively small compared with centrifugal force. So the dynamic aerodynamic force should be focused instead.”

Answer: Uncertainty (unsteadiness of flow and turbulences) is one of the future research subjects.

Quote: „The conclusion with original opinions is not shown in the final part. The statements in the conclusion are common sense descriptions. The authors need to draw more original conclusions.”

Answer: Conclusions were rewritten.

In addition, several other modifications were made (thanks to the other reviews) which also hopefully contributed to improving the quality of the work in question. We hope that the changes made and the clarifications provided will result in the publication: Numerical Investigation of the Influence of Aerodynamic Loads on the Resonant Frequency of a Compressor Blade Made of EI-961 Alloy meeting the requirements of Materials and will be published in this journal.

Kind regards,

Authors.

Reviewer 2 Report

Before going to the comments, let me argue that it seems to me the paper does not exactly suits within the "Materials" journal. A reference to "materials" is just cited in the subject. Of course, any system is made of some material. In my opinion, a way to make the paper closer to the journal subject would be to illustrate "why the test article was made of that material" and "what (beneficial) influence that material can have on the problem at a hand". Of course these are just examples. Many other ways to make the article closer to the target topics. This step is essential.

The paper is interesting, and would deserve better attention.

The bibliography is recent and coherent to the subject.

The introduction should be a bit more extensive... in many cases is written "unfortunately (this is not available)". In the opinion of the authors, how the engineers can overcome that missing info in the everyday design activity? Maybe the solutions are not the optimal, but there are. This is essential to introduce how their work fit into the reduction of the necessary uncertainties we currently face.

Another major aspect concern the experimental validation of the proposed model. Now, the paper deals with an FSI formulation. I understand it is not that easy to get experimental data to validate a certain process. However, something can be done by retrieving some literature data and perform some numerical activity to show the coherence between the achieved results and the reference data. In this moment, I can just say that I saw a FE model of a structure showing a good coherence with the outcome of an experimental modal analysis performed in a lab room. Either, I can just see to have seen the coherence among eigenfrequencies, not even the eigenvectors. Some more effort is necessary from my point of view.

 I think the authors can overcome the issues I pointed out, which were perhaps neglected just for a matter of time. However, a paper cannot limit itself in that way, but it should be more comprehensive, and exhibit a certain solidity of knowledge. Only in this way it can be useful and be used by other This is the only way it can be useful for, and used by other researchers.

Author Response

Dear Reviewer,

In the beginning, I would like to thank you very much for the time spent on reviewing our manuscript and providing suggestions how to improve our publication.

All formatting errors and typos have been corrected. All suggestions were helpful and valuable. The responses and comments to individual remarks will be provided below.

The following structure was adopted in the response: first, the reviewer's remark is quoted, and then the answer is provided.

Quote: „Before going to the comments, let me argue that it seems to me the paper does not exactly suits within the "Materials" journal. A reference to "materials" is just cited in the subject. Of course, any system is made of some material. In my opinion, a way to make the paper closer to the journal subject would be to illustrate "why the test article was made of that material" and "what (beneficial) influence that material can have on the problem at a hand". Of course these are just examples. Many other ways to make the article closer to the target topics. This step is essential.”

Answer: The purpose of the article is not to design the engine but to study the effect of aerodynamic loads on resonant frequencies of a compressor blade made of EI-961 Alloy. This effect is impossible to determine experimentally so the numerical calculation were used. Additionally there are no data on this subject in the literature which was stated in the article. Additionally, an extra statement was added in the Introduction chapter.

Quote: „The introduction should be a bit more extensive... in many cases is written "unfortunately (this is not available)". In the opinion of the authors, how the engineers can overcome that missing info in the everyday design activity? Maybe the solutions are not the optimal, but there are. This is essential to introduce how their work fit into the reduction of the necessary uncertainties we currently face.”

Answer: According to one of the reviewer the introduction is too long already. In the aerospace industry the experimental test are always necessary  for permission to flight which check all the simplifications and assumptions of any calculations (uncertainties). As the development of engines is ongoing and the fact that influence of some parameters are impossible to verify experimentally such calculations as presented in this paper seems to be reasonable. Knowledge that some parameter has small influence is also important knowledge.

Quote: " Another major aspect concern the experimental validation of the proposed model. Now, the paper deals with an FSI formulation. I understand it is not that easy to get experimental data to validate a certain process. However, something can be done by retrieving some literature data and perform some numerical activity to show the coherence between the achieved results and the reference data. In this moment, I can just say that I saw a FE model of a structure showing a good coherence with the outcome of an experimental modal analysis performed in a lab room. Either, I can just see to have seen the coherence among eigenfrequencies, not even the eigenvectors. Some more effort is necessary from my point of view.”

Answer: The presented results are the results that allow the authors to raise the issues discussed in the future. Due to the authors' access to the blades and the EI-961 alloy, they allow for a satisfactory comparison of the experiment and numbers. In the future, wind tunnel tests will be carried out to show the effect of aerodynamic loads.

In addition, several other modifications were made (thanks to the other reviews) which also hopefully contributed to improving the quality of the work in question. We hope that the changes made and the clarifications provided will result in the publication: Numerical Investigation of the Influence of Aerodynamic Loads on the Resonant Frequency of a Compressor Blade Made of EI-961 Alloy meeting the requirements of Materials and will be published in this journal.

Kind regards,

Authors.

Reviewer 3 Report

The paper entitled "Numerical Investigation of the Influence of Aerodynamic Loads on the Resonant Frequency of a Compressor Blade Made of EI-961 Alloy" written by Arkadiusz Bednarz , Krzysztof Puchała, Michał Sałaci ' nski and Volodymyr Hutsaylyuk debates the influence of aerodynamic loads on the value of the resonant frequency of the compressor blade. The authors focused their study on the compressor blade of the 1st stage of the PZL-10W engine. They performed a comparative analysis between experimental results from the literature, results from an analytical model and results from a numerical model based on the FSI. The numerical model is built on a control volume surrounding the blade, taking into account periodicity and inlet and outlet conditions. The authors first perform the aerodynamic study for three compressor speeds, derive the structural loads and then the first three resonance frequency for each speed. The authors draw a number of conclusions regarding the influence of aerodynamic loads.

Overall, the paper is well constructed, based on a substantial bibliographic base. The language is of good quality.

There are almost no spelling or grammatical errors.

There are still minor corrections specified to the authors

Some comments or questions:

- Lines 116 and 117: it is stated in line 116 that for the analytical calculation the authors have taken the cross-section at the foot of the blade. It is not specified for which section moment of inertia was determined (line 117). Is there a big difference between the values at the foot of the blade and the other sections? If so, why not give a range between the minimum and maximum values?

- There is no study of the influence of the mesh size in the case of the CFD calculation. How can the authors guarantee that they have chosen the best mesh size ?

- Concerning the pressure and velocity fields (figures 4 and 5), why did they not present and comment on the results for the three speeds of rotation studied? Why designate the faces as internal or external and not intrados and extrados? Figure 5 should appear in paragraph 3.1 where it is quoted and not in the following paragraph

- It would seem that the calculations were carried out under the assumption of an incompressible fluid. Why not state this explicitly?

- Concerning the boundary conditions, we cannot give conditions that are similar to a flow at both the inlet (98 m/s velocity) and the outlet (201 g/s) because of the conservation of mass equation.

- Figure 3 should be completed with a cross-section along yz to show the prismatic meshes near the walls of the blade.

- As the mesh of the structural calculation is not shown, it is difficult for the reader to understand the shape of the mesh

- What is the relationship between the two meshes and how is the pressure field transmitted from one to the other?

- Line 203 refers to equivalent stress, can the authors specify: von mises, tresca... and why this choice?

- Figure 7: What does the colour scale represent? Can we have a superposition of the undeformed and deformed profiles? Can we have the same type of results for the SP analysis?

Author Response

Dear Reviewer,

In the beginning, I would like to thank you very much for the time spent on reviewing our manuscript and providing suggestions how to improve our publication.

All formatting errors and typos have been corrected. All suggestions were helpful and valuable. The responses and comments to individual remarks will be provided below.

The following structure was adopted in the response: first, the reviewer's remark is quoted, and then the answer is provided.

Quote: „Lines 116 and 117: it is stated in line 116 that for the analytical calculation the authors have taken the cross-section at the foot of the blade. It is not specified for which section moment of inertia was determined (line 117). Is there a big difference between the values at the foot of the blade and the other sections? If so, why not give a range between the minimum and maximum values?”

Answer: The owner of the used geometric model agreed to show only the basic geometric quantities needed for the calculations. Hence, it was necessary to limit the information only to the quantities used. It is obvious that the moment of inertia of individual cross-sections as well as the surface areas change with the location of the cross-section.

Quote: „There is no study of the influence of the mesh size in the case of the CFD calculation. How can the authors guarantee that they have chosen the best mesh size?”

Answer: The selection of the flow grid parameters was made on the basis of a series of simulations, the aim of which was to obtain convergent results (speed distribution at the exit, pressure on the blade surface, etc.). As a result, it was found that the established mesh is suitable for further analyzes.

Quote: „Concerning the pressure and velocity fields (figures 4 and 5), why did they not present and comment on the results for the three speeds of rotation studied? Why designate the faces as internal or external and not intrados and extrados? Figure 5 should appear in paragraph 3.1 where it is quoted and not in the following paragraph”

Answer: These figures present the distributions of selected parameters. They are qualitatively similar for each speed and the selected speed is the higher analyzed, therefore the presented values are maximal. Figure 5 was moved to paragraph 3.1.

Quote: „it would seem that the calculations were carried out under the assumption of an incompressible fluid. Why not state this explicitly?”

Answer: An opinion was added on this. Of couse that was compressible model of air.

Quote: „Concerning the boundary conditions, we cannot give conditions that are similar to a flow at both the inlet (98 m/s velocity) and the outlet (201 g/s) because of the conservation of mass equation.”

Answer: Knowing the velocity parameters at the entry to the stage and thanks to the information on the mass flow at the exit, thanks to the flow continuity equation, the analysis will allow to obtain the velocity at the outflow (which is reflected in the dynamic and total pressure).

Quote: „Figure 3 should be completed with a cross-section along yz to show the prismatic meshes near the walls of the blade.”

Answer: Figure 3 is completed. There is no cross-section – its full view.

Quote: „As the mesh of the structural calculation is not shown, it is difficult for the reader to understand the shape of the mesh”

Answer: Data about the mesh are included in chapter 3.2. The blade discretization is presented in Fig. 6.

Quote: „What is the relationship between the two meshes and how is the pressure field transmitted from one to the other?”

Answer: The mesh in Figure 3 shows the mesh for flow analysis, and the mesh in Figure 6 is for the strength analysis. The pressure distribution from the common surfaces (side surfaces of the blades) is assumed as aerodynamic loads in the strength analysis.

Quote: „Line 203 refers to equivalent stress, can the authors specify: von mises, tresca... and why this choice?”

Answer: The text is completed

Quote: „Figure 7: What does the colour scale represent? Can we have a superposition of the undeformed and deformed profiles? Can we have the same type of results for the SP analysis?”

Answer: The colors represent the theoretical level of deflection on a scale from 0 to 1 mm (displacements scaled up to 1 mm).

In addition, several other modifications were made (thanks to the other reviews) which also hopefully contributed to improving the quality of the work in question. We hope that the changes made and the clarifications provided will result in the publication: Numerical Investigation of the Influence of Aerodynamic Loads on the Resonant Frequency of a Compressor Blade Made of EI-961 Alloy meeting the requirements of Materials and will be published in this journal.

Kind regards,

Authors.

Reviewer 4 Report

The paper deals with an important issue for the operation of a compressor. It certainly has an audience for technicians and researchers in the field. Qualitatively the results seem pertinent, but in what follows I will make some observations to which the authors are asked to clarify. These will be on some issues in terms of the CFD approach.

   1. - Concerning the conditions imposed in the CFD procedure, it is stated that a flow of 201g/s was imposed at the outlet of the calculation domain "the outlet - the outlet was realized with a mass flow of 201 g/s; 162" (page 5) and aerodynamic calculations were made for 3 rotational speeds: 11000, 22490 and 31486 RPM (page 6). As a rule, the mass flow entering the compressor increases with speed. Maintaining the same flow rate when the rpm has increased 3 times seems a bit unrealistic. Explanations are requested on this point.

  2. - "Working medium - air with a density of 1.225 kg/s and other parameters corresponding to the reference atmosphere" (page 6). From this statement it would appear that the authors consider air to be an incompressible gas. The relative velocity distribution in figure 5 shows that the flow regime is compressible. The authors have to specify whether they considered in their simulations that air is considered compressible or incompressible gas.

 3. - From what the authors stated, it would appear that they used the same grid for 3 quite different RPMs 11000, 22490 and 31486 RPM (page 6), i.e. for 3 quite different Reynolds numbers. For this reason information about y+ should be given to allow readers to see if the boundary layer is properly simulated.

4. - on page 7, a rather strange statement is made: "In the absence of rotational speed (the blades were standing), the incoming air itself causes the blade to deflection by nearly 0.004 mm and the appearance of stresses of approximately σeqv = 44 MPa". The authors have to explain why they obtain such a high equivalent stress when the stagnation pressure is 107 207.45 Pa.

 5. - For this reason, the authors have to use several grids, to show that the numerical results do not depend (much) on the grid.

6. - in figure 4, it should be noted that a relative pressure distribution has been plotted. In addition, it must be made very clear whether the plotted pressure distribution is static or total.

7. - in figures 4 and 5, the height of the palette at which the cylindrical section of the calculation range has been made must be specified.

Author Response

Dear Reviewer,

In the beginning, I would like to thank you very much for the time spent on reviewing our manuscript and providing suggestions how to improve our publication.

All formatting errors and typos have been corrected. All suggestions were helpful and valuable. The responses and comments to individual remarks will be provided below.

The following structure was adopted in the response: first, the reviewer's remark is quoted, and then the answer is provided.

Quote: "1. - Concerning the conditions imposed in the CFD procedure, it is stated that a flow of 201g/s was imposed at the outlet of the calculation domain "the outlet - the outlet was realized with a mass flow of 201 g/s; 162" (page 5) and aerodynamic calculations were made for 3 rotational speeds: 11000, 22490 and 31486 RPM (page 6). As a rule, the mass flow entering the compressor increases with speed. Maintaining the same flow rate when the rpm has increased 3 times seems a bit unrealistic. Explanations are requested on this point."

Answer: The presented assumptions of the boundary conditions relate to the conditions of "maximum continuous" operation with a rotational speed of 22,490 rpm. In the case of other calculations, these values were adopted as a linear variable. Explanation / extension of the description was introduced in the content.

Quote: "2. - "Working medium - air with a density of 1.225 kg/s and other parameters corresponding to the reference atmosphere" (page 6). From this statement it would appear that the authors consider air to be an incompressible gas. The relative velocity distribution in figure 5 shows that the flow regime is compressible. The authors have to specify whether they considered in their simulations that air is considered compressible or incompressible gas."

Answer: The analysis is of course based on the compressible model. Specified in the text of the article.

Quote: "3. - From what the authors stated, it would appear that they used the same grid for 3 quite different RPMs 11000, 22490, and 31486 RPM (page 6), i.e. for 3 quite different Reynolds numbers. For this reason, information about y+ should be given to allow readers to see if the boundary layer is properly simulated."

Answer: Unfortunately, the authors did not consider y +. The grid model was determined by a series of parallel interactions until stable results (outlet pressure distribution, mean velocity, etc.) of the analysis were established. In this way, the assumptions for the mesh, common to all simulations, were defined.

Quote: "4. - on page 7, a rather strange statement is made: "In the absence of rotational speed (the blades were standing), the incoming air itself causes the blade to deflection by nearly 0.004 mm and the appearance of stresses of approximately σeqv = 44 MPa". The authors have to explain why they obtain such a high equivalent stress when the stagnation pressure is 107 207.45 Pa."

Answer: Such a case can happen when the engine stops. The whole blade is exposed to pressure connected to the velocity, therefore, the deflection and stresses are the result of all pressure acting (collected by) the blade.

Quote: "5. - For this reason, the authors have to use several grids, to show that the numerical results do not depend (much) on the grid."

Answer: Explained under item 3.

Quote: "6. - in figure 4, it should be noted that a relative pressure distribution has been plotted. In addition, it must be made very clear whether the plotted pressure distribution is static or total."

Answer:  The title of Figure 4 has been changed.

Quote: "7. - in figures 4 and 5, the height of the palette at which the cylindrical section of the calculation range has been made must be specified."

Answer: The selected cross-section is the cross-section that fills the channel in 50% of the height of the blade. Commentary has been added under both figures.

In addition, several other modifications were made (thanks to the other reviews) which also hopefully contributed to improving the quality of the work in question. We hope that the changes made and the clarifications provided will result in the publication: Numerical Investigation of the Influence of Aerodynamic Loads on the Resonant Frequency of a Compressor Blade Made of EI-961 Alloy meeting the requirements of Materials and will be published in this journal.

Kind regards,

Authors.

Reviewer 5 Report

The study determined the influence of aerodynamic loads on the value of the resonance frequency of the first stage compressor blade of the PZL-10W engine. In the calculations, the resonance frequencies were analytically determined and numerically modeled, which was compared with the experiment. As part of the analyzes, both the influence of aerodynamic loads and the rotational speed of the compressor rotor were verified. The paper presented for review is a typical calculation for a compressor blade and after reading it looks like a typical engineering thesis, in which there are no emphasized scientific aspects. In conclusion, the conclusion that aerodynamic loads do not have a large influence on the frequency of resonant vibrations is quite obvious. This was to be expected, especially as it is the first stage of the compressor, with very low pressures and low speeds of the working medium. From a practical point of view, the form of forced vibrations II and III does not matter. Nevertheless, the work is written on a sufficient substantive level, but in my opinion it requires some modifications and additions to be published in the Materials journal. My comments:

1. In the introduction, the ineffective references should be modified. For example, lines 35, 45, and 83 contain multiple references to one sentence. This would require a broader explanation;

2. In my opinion, the analysis of the influence of the aerodynamic load in rotating machines makes sense when we analyze the entire stage and not just a single blade. There is no interaction of other blades and other stages. In this situation, analyzing the first degree, although it is the longest, is the least interesting from the scientific point of view.

3. The discrepancy between the calculations and the experiment for modes II and III of forced vibrations also requires a broader explanation, not only the statement of this fact.

4. In my opinion, the summary also requires expansion or corrections, with an emphasis on scientific achievements.

5. The list of bibliographies requires corrections to be adjusted to the guidelines of the journal.

Author Response

Dear Reviewer,

In the beginning, I would like to thank you very much for the time spent on reviewing our manuscript and providing suggestions how to improve our publication.

All formatting errors and typos have been corrected. All suggestions were helpful and valuable. The responses and comments to individual remarks will be provided below.

The following structure was adopted in the response: first, the reviewer's remark is quoted, and then the answer is provided.

Quote: „1. In the introduction, the ineffective references should be modified. For example, lines 35, 45, and 83 contain multiple references to one sentence. This would require a broader explanation”

Answer: The references are modified. Items from line 83 were presented in the previous chapter. In line 83, the reference was made to the strength data of the EI-961 material and the ongoing fatigue tests. The description has been developed for the described items.

Quote: „ 2. In my opinion, the analysis of the influence of the aerodynamic load in rotating machines makes sense when we analyze the entire stage and not just a single blade. There is no interaction of other blades and other stages. In this situation, analyzing the first degree, although it is the longest, is the least interesting from the scientific point of view.”

Answer: No necessarily. In reality, each blade has slightly different modal modes because of errors that always occur when any object is produced. Therefore analysis of a single blade is also important to the subject. Of course, this is only a step of an analysis of the entire engine. The analyzes assumed symmetry (continuity) between the domains, so the theoretical mutual influence of successive blades was also taken into account.

Quote: „ 3. The discrepancy between the calculations and the experiment for modes II and III of forced vibrations also requires a broader explanation, not only the statement of this fact. ”

Answer: The text was modified.

Quote: „ 4. In my opinion, the summary also requires expansion or corrections, with an emphasis on scientific achievements. ”

Answer: Conclusions were rewritten.

Quote: „ 5. The list of bibliographies requires corrections to be adjusted to the guidelines of the journal.”

Answer: The publication was prepared in the Latex language, using the built-in formatting for the bibliography (prepared by the MDPI publisher).

In addition, several other modifications were made (thanks to the other reviews) which also hopefully contributed to improving the quality of the work in question. We hope that the changes made and the clarifications provided will result in the publication: Numerical Investigation of the Influence of Aerodynamic Loads on the Resonant Frequency of a Compressor Blade Made of EI-961 Alloy meeting the requirements of Materials and will be published in this journal.

Kind regards,

Authors.

Round 2

Reviewer 4 Report

The answers of the authors have brought clarifications in the proportion of 70-80%. On the whole, the paper does, however, solve a problem of interest.

Author Response

Thanks for the information, I'm glad it cleared things up a bit. Undoubtedly, any comments obtained during the review will help to raise the level of science in all future publishing activities.
Kind regards,
Arkadiusz Bednarz